# An Exception to the Rule? Could Photobiont Identity Be a Better Predictor of Lichen Phenotype than Mycobiont Identity?

**DOI:** 10.3390/jof8030275

**Published:** 2022-03-09

**Authors:** Jana Steinová, Håkon Holien, Alica Košuthová, Pavel Škaloud

**Affiliations:** 1Department of Botany, Faculty of Science, Charles University in Prague, Benátská 2, 128 01 Prague, Czech Republic; skaloud@natur.cuni.cz; 2Faculty of Biosciences and Aquaculture, Nord University, Pb 2501, NO-7729 Steinkjer, Norway; hakon.holien@nord.no; 3Department of Botany, Swedish Museum of Natural History, P.O. Box 50007, SE-104 05 Stockholm, Sweden; kosuthova.alica@gmail.com

**Keywords:** *Asterochloris*, barcoding, *Cladonia*, lichens, speciation, species delimitation

## Abstract

With rare exceptions, the shape and appearance of lichen thalli are determined by the fungal partner; thus, mycobiont identity is normally used for lichen identification. However, it has repeatedly been shown in recent decades that phenotypic data often does not correspond with fungal gene evolution. Here, we report such a case in a three-species complex of red-fruited *Cladonia* lichens, two of which clearly differ morphologically, chemically, ecologically and in distribution range. We analysed 64 specimens of *C. bellidiflora*, *C. polydactyla* and *C. umbricola*, mainly collected in Europe, using five variable mycobiont-specific and two photobiont-specific molecular markers. All mycobiont markers exhibited very low variability and failed to separate the species. In comparison, photobiont identity corresponded better with lichen phenotype and separated esorediate *C. bellidiflora* from the two sorediate taxa. These results can be interpreted either as an unusual case of lichen photomorphs or as an example of recent speciation, in which phenotypic differentiation precedes the separation of the molecular markers. We hypothesise that association with different photobionts, which is probably related to habitat differentiation, may have triggered speciation in the mycobiont species.

## 1. Introduction

In their latest definition [1], lichens are characterised as “self-sustaining ecosystems formed by the interaction of an exhabitant fungus and an extracellular arrangement of one or more photosynthetic partners and an indeterminate number of other microscopic organisms”. Over the last decade, the importance and contribution of bacteria, yeasts, and microscopic fungi has also been recognised, though these organisms are probably less integrated into the symbiotic network than the more tight and intimate relationship between the exhabitant fungus (the mycobiont) and its photosynthetic partners (photobiont) [1,2,3]. From the fungal perspective, lichenization represents one of the most successful nutritional strategies, with around 20% of fungal species lichenised [4].

Lichens display several unique adaptations that separate them from other symbiotic associations. In addition to their impressive longevity and endurance (extreme tolerance of drought, heat and cold stress), lichens are especially fascinating in the way their thallus is constructed. If lichen photobionts and mycobionts are cultured separately, for example, they will have different body forms than the symbiotic thallus composed of both partners [5,6], unlike other symbiotic associations, such as corals, which have the shape and form of the thallus (body) of just one of the participants in the symbiotic association [3].

Lichen thalli have a broad spectrum of forms, most of which are relatively simple in their organisation and are often inconspicuous (crustose, microfilamentous or microglobose thalli). However, a relatively low number (ca. 25%) of lichenised fungi form morphologically and anatomically complex leaf- or shrub-like symbiotic phenotypes, i.e., the so-called macrolichens [5].

It has traditionally been assumed that the lichen phenotype is determined by the primary mycobiont [3], and this remains the prevalent view, despite recent findings that have challenged this concept [7], with Lücking et al. [6], for example, stating that “mycobionts build a greenhouse for the photobiont”. The critical role of primary mycobionts on phenotype expression is also reflected in its use for the scientific name of lichens [3,6]. While the numbers of lichens with phenotypes determined by a partner other than the primary mycobiont is relatively low, recent studies have highlighted a possible role of cortical biofilms containing basidiomycete yeasts in determining lichen phenotype in some macrolichens [7,8]. However, the most common examples of this phenomenon to date are the photosymbiodemes, in which the same fungus forms different structures, or entirely different lichens, when associating with either green algae or cyanobacteria, e.g., [9,10,11].

Until the 1990s, species of lichenised fungi were defined solely based on phenotypic traits (i.e., morphological, anatomical and chemical features); however, in recent years, scientists have obtained new tools for discriminating species with the introduction of molecular phylogenetic methods. The use of molecular data has several advantages; first, it provides a conceptually independent framework for defining species (unless the markers employed are correlated with diagnostic phenotypic characteristics), and second, the number of characteristics that can be used to define lineages is much higher compared to phenotypic characteristics [6]. Despite the clear benefits of such DNA-based methods, taxonomists often face serious problems in situations where phenotypic data do not correspond with fungal gene evolution [8,12,13,14].

During our previous work, we observed that two distinctive *Cladonia* species (*C. bellidiflora* and *C. polydactyla*) that clearly differed morphologically, chemically [15], ecologically and in distribution range could not be successfully distinguished by their fungal ITS rDNA sequences (i.e., recent mycobiont barcoding markers). We then decided to focus on this striking phenomenon, examining more material and making use of both mycobiont- and photobiont-specific molecular markers. A third species (*C. umbricola*), close to *C. polydactyla* [15,16,17], was also included in the study to give a wider view. In the present study, we assessed (1) whether it is possible to distinguish taxa that clearly differ phenotypically using any of the commonly used fungal molecular markers, and (2) which factor among such lichens best explains the different phenotypes (traditionally attributed to different mycobiont species). In answering these questions, we aimed to provide a more complex understanding of aspects influencing the establishment of symbiotic lichen phenotypes and the evolution of lichen-forming fungi.

## 2. Materials and Methods

### 2.1. Taxon Sampling and Determination

In total, 64 specimens of *C. bellidiflora*, *C. polydactyla* and *C. umbricola* (Figure 1) were sampled, covering the widest range of morphological, ecological, geographical and chemical diversity (Table 1) as possible. Thirty specimens (ten of each species), sampled from the Czech Republic, Norway, Scotland, Sweden, Switzerland, Wales and the Canary Islands (Figure 2), represented the main dataset, which was then subjected to detailed study using seven genetic markers (see Section 2.2). For the remaining 34 specimens (16 × *C. bellidiflora*, 11 × *C. polydactyla*, 4 × *C. umbricola*; herein termed the “extended dataset”), also collected mainly from Europe, we amplified fungal and/or algal ITS rDNA (detailed information for all specimens used in this study are listed in Appendix A). Lichens were determined to species by using both morphological and chemical characteristics, while secondary metabolites were identified using thin-layer chromatography (TLC) on Merck silica gel 60 F254 pre-coated glass plates in solvent systems A and C, according to Orange et al. [18].

### 2.2. Selection of Genetic Markers

The molecular markers used for mycobiont molecular identification were selected based on the results of previous studies focusing on the genus *Cladonia* [19,20,21,22]. Markers with sufficient resolution, i.e., those that were previously successful in distinguishing *Cladonia* species or showed high variability, were used alongside the more traditional *Cladonia* barcode marker ITS rDNA (e.g., [19]). The second largest subunit of the RNA polymerase II gene (RPB2), which showed the highest percentage of correct *Cladonia* species identification, was proposed as a candidate for the second *Cladonia* barcode marker by Pino-Bodas et al. [19]. In the same study, mitochondrial cytochrome c oxidase I (cox1) was also suggested as a candidate barcode marker as it was the only marker shown to have a barcoding gap [19]. Cox1 has also been shown to display high infraspecific variation [22]. Elongation factor-1α (EF-1α), also proposed as an additional barcode marker for fungi [21], is commonly used for *Cladonia* [20,23,24]. Finally, Kanz et al. [20] demonstrated that the small subunit of mitochondrial ribosomal DNA (mtSSU) had the highest discriminatory power to distinguish *Cladonia* species. 

In addition to the above-mentioned mycobiont primers, we also amplified the algal ITS rRNA, which is commonly used for the photobionts of *Cladonia* lichens (e.g., [25,26,27]). For selected samples, we also amplified a photobiont actin type I locus in order to classify *Asterochloris* sequences to species level [28]. 

### 2.3. DNA Extraction, PCR and Sequencing

Dried lichen material was carefully checked under a dissecting microscope to ensure a lack of contamination from other lichens, after which it was homogenised and used for total DNA extraction following the CTAB protocol [29]. Six molecular markers were amplified using the following primers: ITS1F [30] and ITS4 [31] for fungal ITS rDNA, CLRPB2-5F and CLRPB2-7R [24] for RPB2, mrSSU1 [32] and MSU7 [33] for mtSSU, cox1-5959F and cox1-6711R [22] for the cox1 gene, CLEF-3F and CLEF-3R [24] for part of the elongation factor-1α, ITS1T and ITS4T [34] for algal ITS rDNA and actin_F and actin_R [35] for the photobiont actin type I locus. PCR reactions were performed using MyTaq polymerase (Bioline, London, UK) in 20 µL volume (PCR conditions summarised in Appendix A). The PCR products were visualised on a 1% agarose gel stained with ethidium bromide and subsequently purified using the Agencourt Ampure XP system (Beckman Coulter, Brea, CA, USA). Sequencing was performed at Macrogen Inc., using an ABI 3730 DNA analyser (Applied Biosystems, Waltham, MA, USA) and the same primers used for PCR amplifications. 

### 2.4. Sequence Alignment and Phylogenetic Analysis

All sequences obtained were checked and assembled using the software package SeqAssem [36]. Additional mycobiont and photobiont sequences were downloaded from GenBank (https://www.ncbi.nlm.nih.gov/ (accessed on 25 November 2021)). (All sequences included in the dataset are summarised in Appendix A). Alignments were either manually built in MEGA7 [37] or constructed using MAFFT version 6 [38] under the QINS-I strategy. For the actin gene, we used Gblocks to remove introns from the alignment and to eliminate poorly aligned positions [39]. The final datasets comprised 552 nucleotide sites of mycobiont ITS rDNA, 797 sites of RPB2, 585 sites of EF-1α, 709 sites of coxI, 903 sites of mtSSU, 461 sites of photobiont ITS rDNA and 632 sites of the photobiont actin type I gene. Information on the number of variable sites, number of parsimony informative sites, intraspecific and interspecific distances were retrieved using the MEGA7 software package [37].

Genealogical relationships for mycobiont ITS rDNA, RPB2, EF-1α, coxI, mtSSU and photobiont ITS rDNA were investigated by constructing maximum parsimony (MP) haplotype networks using the Haplotype Viewer (Ewing; available at http://www.cibiv.at/~greg/haploviewer (accessed on 25 November 2021)).

To classify the newly obtained algal sequences, we produced phylogenetic trees from the concatenated photobiont dataset using the Bayesian inference (BI), maximum likelihood (ML) and weighted maximum parsimony (wMP) approaches. Nucleotide-substitution models were selected independently for both photobiont loci (ITS rRNA and the actin gene) according to the Bayesian information criterion (BIC), as implemented in jModelTest 2.1.6 [40]. We applied the TIM2ef+G model for the photobiont ITS1 and ITS2 partitions, the JC model for the 5.8S partition, and the TPM2uf+I+G model for the actin gene. MrBayes version 3.2.7 [41] was used to construct the phylogenetic tree, using two parallel MCMC runs with four chains for 15 million generations, with trees and parameters sampled every 100 generations. Convergence of the chains was assessed during the run by calculating the average standard deviation of split frequencies (SDSF), the SDSF between simultaneous runs being >0.01. Burn-in values were determined using the “sump” command, which discards 25% of initial trees. Bootstrap analyses were performed by ML using RAxML version 8.2.10 [42] and MP analyses using PAUP version 4.0b10 [43]. ML analysis consisted of 100 tree replicates and 1000 rapid bootstrap inferences with automatic termination. As the tree topology obtained using the ML method agreed with the Bayesian tree topology, the Bayesian phylogram only is shown herein. The MP analysis was performed using heuristic searches with 1000 random sequence addition replicates and random addition of sequences, with MP bootstrap support values obtained using 1000 bootstrap replicates. The resulting trees were visualised using FigTree version 1.4.4 [44].

### 2.5. Variation Partitioning

Variation partitioning analyses were performed in R 4.0.5 (R Core Team, 2020) using base functions and the packages ape [45], Geiger [46], geosphere [47], phytools [48], SoDA [49] and vegan [50]. We first evaluated the relative effects of photobiont diversity, climate, geography and substrate on variance in mycobiont genetic diversity [51]. Subsequently, information on mycobiont and photobiont genetic diversity was transferred to the phylogenetic distances from ML trees and transformed into principal coordinate analysis (PCoA) axes. Geographical distances (latitude and longitude) were then transformed to the principal coordinates of neighbour matrices (PCNM) vectors, which represent geographical distances at different spatial scales [52]. The substrate from which the samples were collected was used as an explanatory variable, alongside data for 19 bioclimatic variables retrieved from the WorldClim database [53]. To select explanatory variables for inclusion in the variation partitioning analysis, we first transformed the variables into principal component analysis (PCA) axes, then selected the most important axes using the broken-stick distribution [54] via the bstick function. The significance of the explanatory variables was then tested using redundancy analysis (RDA). 

## 3. Results

The final dataset included 241 sequences, of which 205 were newly produced (51 sequences of mycobiont ITS rDNA, 30 of cox1, 29 of EF-1α, 27 of mtSSU, 32 of RPB2, 33 sequences of photobiont ITS rDNA and three of photobiont actin type I gene), 15 were retrieved from our previous datasets [55,56] and 21 downloaded from GenBank (https://www.ncbi.nlm.nih.gov/ (accessed on 25 November 2021)). The newly obtained sequences have been deposited in GenBank (see Appendix A for accession numbers). Sequence data were clear and unambiguous, indicating that only single genotypes of both mycobionts and photobionts were present in the thallus.

### 3.1. Mycobiont and Photobiont Diversity

Haplotype network analysis indicated a surprisingly low level of genetic variation in the mycobiont data for the three species. Interestingly, none of the haplotype network structures for the mycobiont genes examined were correlated with lichen species identity based on phenotypic determination (Figure 3). 

Mycobiont ITS rDNA from the 69 samples contained 14 haplotypes, each of which differed by one to four mutational steps. The largest haplotype comprised 23 samples belonging to all three species examined. The cox1 gene haplotype network contained six haplotypes that were separated by one or two mutations, the largest of which contained 16 samples for all three species. The mtSSU dataset harboured five haplotypes that differed from each other by a single mutational step, the largest of which consisted of 14 sequences belonging to *C. polydactyla* and *C. umbricola*. The EF-1α and RPB2 genes both contained three haplotypes, with those for EF-1α separated by a single mutational step and those for RPB2 separated by one to three mutational steps. The largest EF-1α haplotype contained 19 sequences belonging to the sorediate species *C. polydactyla* and *C. umbricola*, while the second-largest haplotype consisted of 10 sequences belonging to *C. bellidiflora* and *C. polydactyla*. The largest RPB2 haplotype comprised 24 sequences for all three species, and the other two haplotypes, each separated by two mutations, contained *C. bellidiflora* sequences only. 

Photobiont ITS rDNA haplotype network (Figure 4) consisting of 17 unique haplotypes can be divided into two main subgroups. The first subgroup harboured six haplotypes containing sequences belonging to *Cladonia bellidiflora* exclusively. The other subgroup contained nine haplotypes and consisted of sequences of both sorediate species (*C. polydactyla* and *C. umbricola*) species, which were intermixed.

The topology of the phylogenetical tree obtained through Bayesian analysis of the photobiont concatenated ITS rDNA and actin dataset (Figure 5) was in agreement with previously published *Asterochloris* phylogenies [26,57,58]. Photobionts associated with the *Cladonia* species were classified to nine *Asterochloris* lineages, of which five represented formally described species, three belonged to undescribed species and one (J2) was positioned on an individual branch close to *A. magna*. All samples of *C. bellidiflora* (*n* = 17) contained photobionts belonging to a strongly supported monophyletic clade composed of *A. glomerata* (five sequences), *A. irregularis* (six sequences), *A. pseudoirregularis* (five sequences) and a single unassigned *Asterochloris* sequence. *Cladonia polydactyla* (16 samples) was associated with *A. magna* (five sequences), *A. italiana* (six sequences), *A.* aff. *italiana* (three sequences), clade I2 (one sequence) and a single unassigned *Asterochloris* sequence (related to *A. magna*). Mycobionts of *C. umbricola* were associated with photobionts belonging to *A. magna* (seven sequences), *A. italiana* (one sequence) and a clade A11 (one sequence). All lineages including more photobionts of the *Cladonia* taxa comprised samples from different geographic areas. 

The molecular markers used in this study varied in resolution (Table 2), with ITS rDNA having the highest number of variable sites (14) and parsimony informative sites (seven) when evaluating mycobiont-specific markers. Cox1 had five variable and parsimony-informative sites, RPB2 four, mtSSU three and EF-1α just two variable and parsimony-informative sites. When comparing the inter-/intraspecific distances of individual mycobiont-specific molecular markers, RPB2 achieved the highest value (4.77) and cox1 the lowest (1.28).

### 3.2. Variation Partitioning

Variation partitioning analysis using photobiont genetic distance, climate and substrate as explanatory variables explained 79% of variability in the genetic distance of mycobionts (Figure 6). Geography was strongly correlated with climate, and hence was removed as an explanatory variable. 

Photobiont genetic distance had the highest explanatory power, explaining 56% of total variability, of which 13% accounted for independent effect, 40% in combination with substrate and 3% in combination with substrate and climate. Substrate explained 54% of total variability, with an independent effect of 11%, while climate explained 20% of total variability with a net effect of 17%.

## 4. Discussion

In this work, we studied species delimitation in three closely-related red-fruited *Cladonia* species: *C. bellidiflora*, *C. polydactyla,* and *C. umbricola*. Despite their obvious morphological and chemical differences [15], *C. bellidiflora* displayed a surprising degree of phylogenetic closeness to the two sorediate species, a feature also reported by Stenroos et al. [23]. In contrast, the status of *C. umbricola* toward *C. polydactyla* has been uncertain for some time [59,60], with some authors simply treating *C. umbricola* as a variant of *C. polydactyla* [61]. The fact that these two sorediate taxa could not be separated by any of the markers used in this study suggests that the two taxa probably represent a single species; confirming some previous taxonomic assumptions and observations on phenotypic variation. Hereon in, therefore, we refer to these taxa as a single sorediate taxonomical entity (*C. polydactyla*/*umbricola*).

Our results demonstrate a case whereby lichens that differ in morphology, reproduction strategy, chemistry, ecology and distributional range could not be distinguished by any of the five mycobiont molecular markers used (of which two are proposed as barcode markers and three have previously been used successfully for species delimitation studies) but differed in their associated photobionts. In all cases, esorediate *C. bellidiflora* samples were associated with different photobiont lineages compared to sorediate *C. polydactyla*/*umbricola* and, as such, our results highlight one of the few known cases in lichen symbiosis where photobiont identity better explains lichen thallus phenotype than mycobiont identity.

The term photomorphs (also photosymbiodemes or photopairs) refers to the situation where the same lichenised fungal species forms different morphotypes depending on the associated photobiont species [10,11,62]. In most cases, these different photomorphs contain either green algae (chloromorphs) or cyanobacteria (cyanomorphs). In the best-known representative of this phenomenon, the cyanomorphs are characterised by a very distinctive coralloid morphology that clearly differs from the foliose thalli of chloromorphs [10]. These were described as a separate genus (*Dendriscocaulon*) as early as the 19th century [63]. Another striking example is the case of *Buellia violaceofusca* and *Lecanographa amylacea,* which were previously synonymised [64]. These taxa have identical nuITS and intermixed mtSSU sequences but contain phylogenetically distant photobionts (*Trebouxia* vs. trentepohlioid). 

Though the signal in our data indicates a key role for photobionts in the phenotypic distinction of this group of lichens, the pattern requires cautious interpretation. While it is possible that *C. bellidiflora* and sorediate *C. polydactyla*/*umbricola* represent different photomorphs of the same fungal species, this is not the only plausible explanation. Chloromorphs, as previously described, are characterised by their association with distant photobiont lineages (i.e., green algae vs. cyanobacteria, *Trebouxia* vs. trentepohlioid algae) [10,64] and, to our knowledge, such associations have never been reported for closely-related photobiont taxa. Furthermore, the sorediate *C. polydactyla*/*umbricola* in this study did not associate with a single monophyletic algal lineage (in contrast to *C. bellidiflora*) but with a higher number of non-monophyletic *Asterochloris* lineages.

An alternative explanation for the observed pattern may be that esorediate *C. bellidiflora* and sorediate *C. polydactyla* (including *C. umbricola*) may represent young diverging species whose molecular markers have not yet been sorted, though the lichens have already differentiated phenotypically. This interpretation is consistent with the “general (unified) lineage concept” (GLC) [65,66,67,68], which defines species as “segments of separately evolving metapopulation lineages” and was proposed as a practical solution to the species concept impasse for lichenised fungi [66]. The GLC is based on the assumption that different properties separating lineages arise at different times during the speciation process, and that they do not necessarily occur in a regular order [68]. In practice, this means that differentiation of phenotypic features may have preceded the separation of molecular markers used in this study. The phenomenon of incomplete lineage sorting has been reported repeatedly from lichenized fungi, e.g., [69,70,71].

Interestingly, the genus *Cladonia*, as with many other macrolichen genera (e.g., *Bryoria*, *Usnea*), is notorious for problems with species delimitation, with molecular data frequently not corroborating with traditional phenotypic species definitions based on morphological and/or chemical characteristics [23,72,73,74,75,76]. Stenroos et al. [23], in their worldwide study of 304 Cladoniaceae species, attributed low genetic differentiation and poorly resolved phylogenies in several *Cladonia* groups to the recent divergence of species and proposed genome-level studies sampling a large number of markers as a possible solution. Just such an approach, based on collecting a higher number of loci, has already proved suitable in another tricky group of lichens, the genus *Usnea*, with RAD sequencing and the use of microsatellite markers confirming the separation of sorediate *U. antarctica* and apotheciate *U. aurantia-coatra* [77,78], both of which differ morphologically but proved genetically inseparable in previous molecular multi-locus studies [12,79]. Since the application of such a robust methodology could clearly shed light onto similar complicated taxonomic puzzles, we refrain from making any taxonomic conclusions in this study and instead wait for results with a higher resolution power (i.e., optimal genomic data).

It is important to mention that though *C. bellidiflora* clearly differs morphologically, chemically and ecologically from the two sorediate taxa in Europe, the situation in other parts of the world may not be so simple. For example, the consistent chemical pattern for *C. bellidiflora* (i.e., presence of usnic and squamatic acids) typical for Europe has been shown to be occasionally replaced by another chemotype containing thamnolic or fumarprotocetraric acid in some areas of North and South America [16,59,80]. At the same time, the presence of thamnolic acid is also characteristic for *C. polydactyla*. Squamatic and thamnolic acids are β-orcinol depsides, which are structurally similar to one another, and the production of these compounds in the same or closely related species is also relatively common in other lichen genera, such as *Siphula, Usnea* or *Thamnolia* [81]. Remarkably, though there were no problems distinguishing *C. bellidiflora* and the two sorediate species, there is a North American endemic species (*C. transcendens*) that can be confused with both *C. bellidiflora* and *C. umbricola* [59]. *Cladonia transcendens* can produce both soredia and squamules and, as such, could potentially represent a transition between two extreme morphological (sorediate and squamulose) states. Unfortunately, mycobiont and photobiont sequences, which would point to its taxonomic position, are presently unavailable for this species. Nevertheless, the high chemical and morphological variability displayed by this North American species complex indicates that this region may be a centre of diversification in this lichen group. As such, it is desirable that material from this area is included in future studies.

Whether or not *C. bellidiflora* and *C. polydactyla/umbricola* represent a single mycobiont species, or two diverging or recently diverged mycobiont species, the correlation of different *Cladonia* phenotypes with two differentiated photobiont pools is clear. A specific association of the lichenised fungus with one or another pool of photobionts may be one of the forces that drive mycobiont speciation, and is likely to go hand-in-hand with niche differentiation [82,83,84]. This corresponds well with the different habitats in which *C. bellidiflora* and *C. polydactyla/umbricola* grow, as well as the fact that photobiont and substrate were the most relevant factors explaining the genetic distance of the mycobiont (most of this effect was shared at 40% out of 54% and 56%, respectively). Characteristics of habitats in which *C. bellidiflora* and *C. polydactyla/umbricola* grow correspond with the known ecological preferences of particular *Asterochloris* species, where such information is available. For example, *C. polydactyla/umbricola,* which commonly grow on dead wood among bryophytes in humid forests, is associated with *A. italiana*, which has previously been reported from *Stereocaulon* and *Cladonia* spp. in relatively humid and stable climates [55,58]. Similarly, *A. irregularis*, which is restricted to cold areas of the northern hemisphere [26], was one of the associated photobionts of *C. bellidiflora*, which usually occurs on mineral soils or boulders in tundra, mountain heaths or on higher hilltops. In such cases, morphological differentiation may then reflect adaptations to specific ecological niches. The podetia of *C. bellidiflora* are strengthened by squamules and a cortex that provide protection against the harsh conditions in open cold habitats. In contrast, the slender podetia of C. *polydactyla/umbricola* is covered with soredia and the cortex is limited, which probably represents an adaptation to the more stable and humid conditions found on bryophyte-covered trunks in coniferous forests. 

Previous studies have shown that sorediate podetia (including those of *C. polydactyla*) are significantly more hydrophobic than corticate podetia [85], most likely as retention of water in stable humid environments is either not necessary or may even be disadvantageous due to the risk of thallus supersaturation [85]. Similarly, it has been reported that soredia attach to water drops on the podetium surface of another sorediate *Cladonia* species, and that bouncing drops carry soredia and disperse them [86]. Interestingly, Škvorová et al. [56] found a negative association between the occurrence of *A. italiana* and the production of soredia in different *Cladonia* species, suggesting that a stable and humid climate might be disadvantageous for sorediate *Cladonia* species. The authors further found a strong positive association between soredia production and the occurrence of *A. glomerata*. This assumption also corresponds to the results of our previous work, in which we found a correlation between photobiont identity and the lichen reproduction mode in another group of red-fruited *Cladonia* lichens [87]. In the case of zeorin-containing *Cladonia* lichens, sorediate species tend to associate with *A. glomerata* and *A. irregularis* exclusively, whereas the esorediate species associate mainly (but not exclusively) with other *Asterochloris* lineages, including *A. italiana*. However, our most recent results contradict this pattern and point to the need for further study of this phenomenon. 

Another aspect in which our results question our previous hypothesis is the level of mycobiont specificity in relation to lichen reproduction mode. Previously, we hypothesised that the dominant asexual reproduction mode (secured by the ability to produce soredia that carry both the mycobiont and photobiont) leads to a more specific and well-tuned relationship between both partners; whereas esorediate species, while dispersing by fungal spores, tend to be less specific toward the algal partner since they always need to establish symbiosis *de novo* from the pool of locally available photobionts. A similar pattern was also found in other studies, whether focused on the genus *Cladonia* [26] or on other lichen groups [88,89]. Clearly, this hypothesis does not apply in the case of *C. bellidiflora*, *C. polydactyla* and *C. umbricola*, where we observe the exact opposite trend, with mainly sexually reproducing species being more specific toward the photobionts than sorediate species. Our new results indicate that the ability to produce soredia is indeed somehow linked with photobiont diversity and specificity; however, this relationship is not necessarily straightforward.

## 5. Conclusions

In this study, we were unable to separate a three-species complex of red-fruited *Cladonia* lichens (*C. bellidiflora*, *C. polydactyla* and *C. umbricola*) despite using five fungal molecular markers (ITS rDNA, RPB2, EF-1α, mtSSU and cox1). We detected a surprisingly low level of mycobiont sequence genetic differentiation in all the molecular markers studied. We found that photobiont identity corresponded better with phenotypic features distinguishing squamulose *C. bellidiflora* and sorediate *C. polydactyla*/*C. umbricola* than mycobiont data. These results can be interpreted either as an unusual case of lichen photomorphs, where the same mycobiont forms different morphotypes according to the associated photobionts, or as an example of recent speciation in which phenotypic differentiation precedes the separation of the molecular markers. In the second case, we hypothesise that it may be the association with different photobionts that triggers speciation of mycobiont species, which is likely related to niche differentiation. Interestingly, sorediate taxa, in which both partners are dispersed together and are expected to show a high level of reciprocal specialisation, displayed a lower level of specificity toward their symbiotic partners than esorediate species.

## Figures and Tables

**Figure 1 jof-08-00275-f001:**
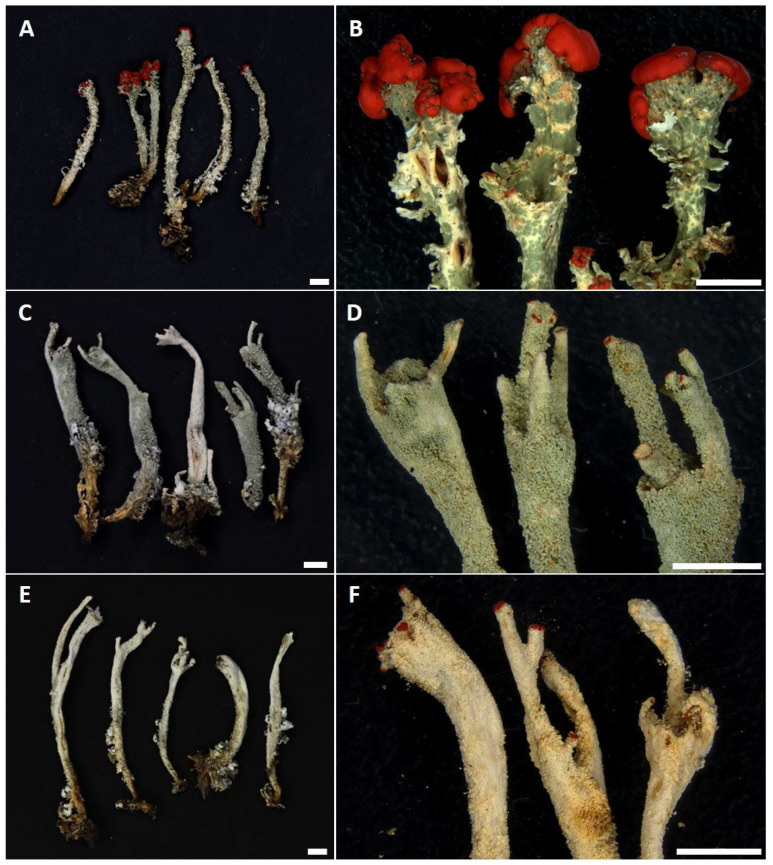
Morphology of the *Cladonia* species studied. (**A**,**B**) = *Cladonia bellidiflora* (J13); (**C**,**D**) = *Cladonia polydactyla* (WBN8A); (**E**,**F**) = *Cladonia umbricola* (A11). Scale bar = 5 mm.

**Figure 2 jof-08-00275-f002:**
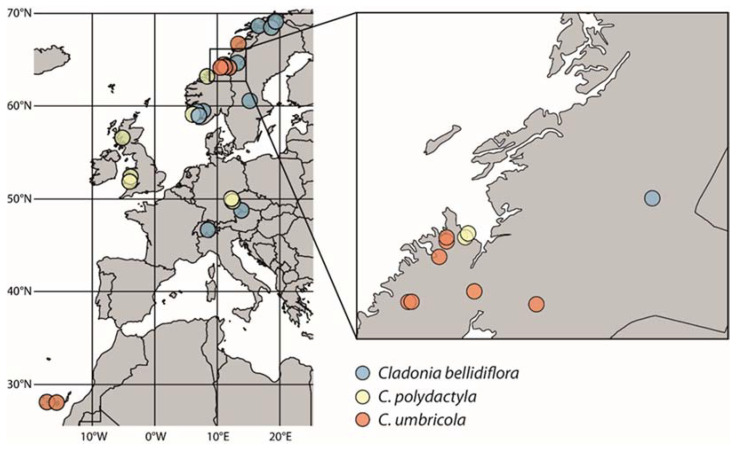
Geographic location of collections used in the main dataset.

**Figure 3 jof-08-00275-f003:**
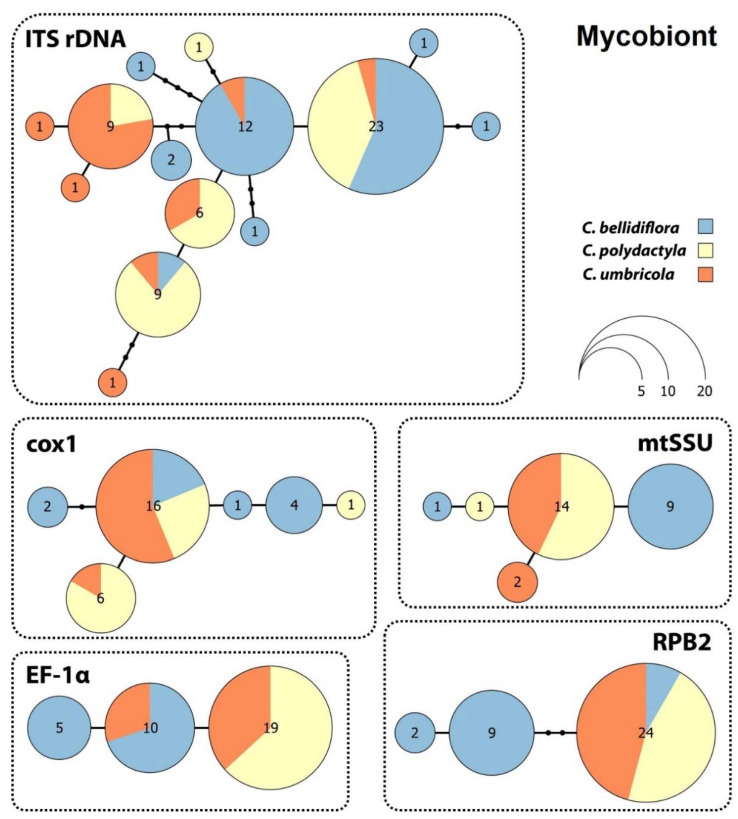
Haplotype networks for *Cladonia bellidiflora*, *C. polydactyla* and *C. umbricola*, based on five fungal molecular markers.

**Figure 4 jof-08-00275-f004:**
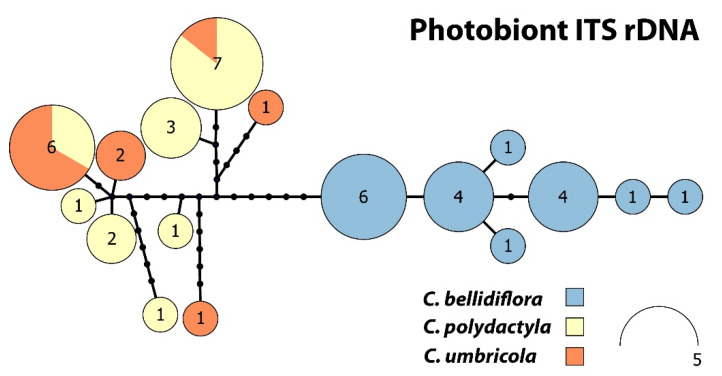
Haplotype network for *Cladonia bellidiflora*, *C. polydactyla* and *C. umbricola*, based on photobiont ITS rDNA.

**Figure 5 jof-08-00275-f005:**
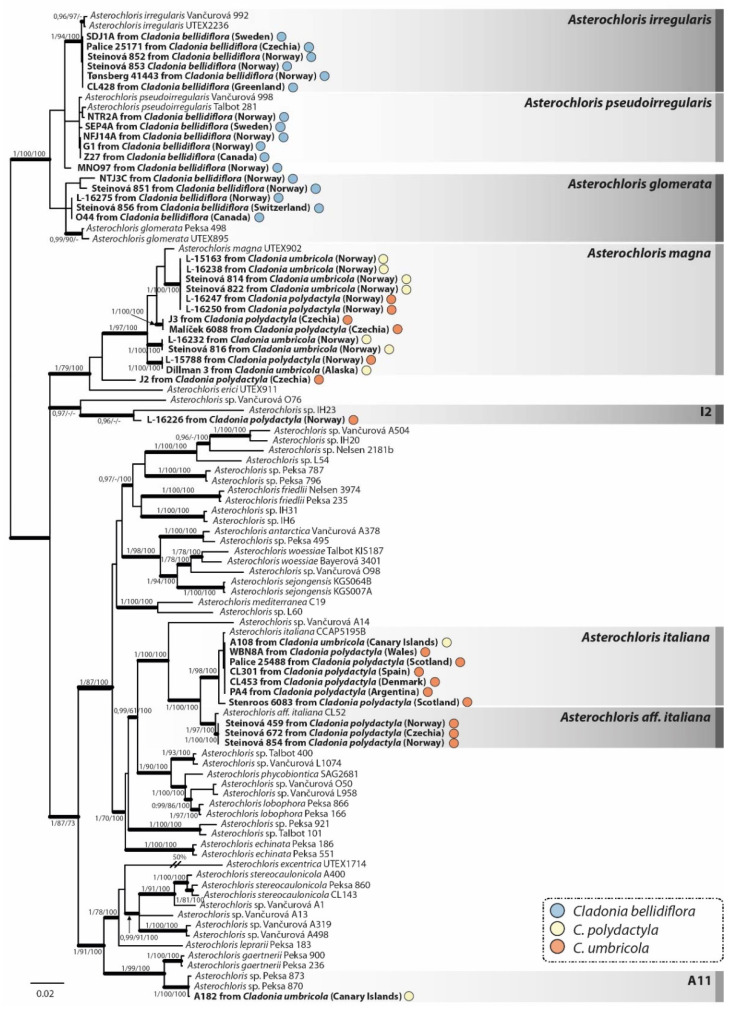
Phylogenetic Bayesian inference of *Asterochloris* photobionts, based on the combined dataset of ITS rDNA and actin I loci. Bayesian posterior probability values (left) and bootstrap support for the ML (middle) and MP (right) analyses are reported at the corresponding branches (only values >0.95 shown for PP, and >70 for bootstrap). The colour of the dots indicates the mycobiont species from which the photobiont originated, i.e., blue = *Cladonia bellidiflora*, orange = *C. polydactyla* and yellow = *C. umbricola*. *Asterochloris* lineages associated with any of the *Cladonia* species studied are in bold. The scale bar shows the estimated number of substitutions per site.

**Figure 6 jof-08-00275-f006:**
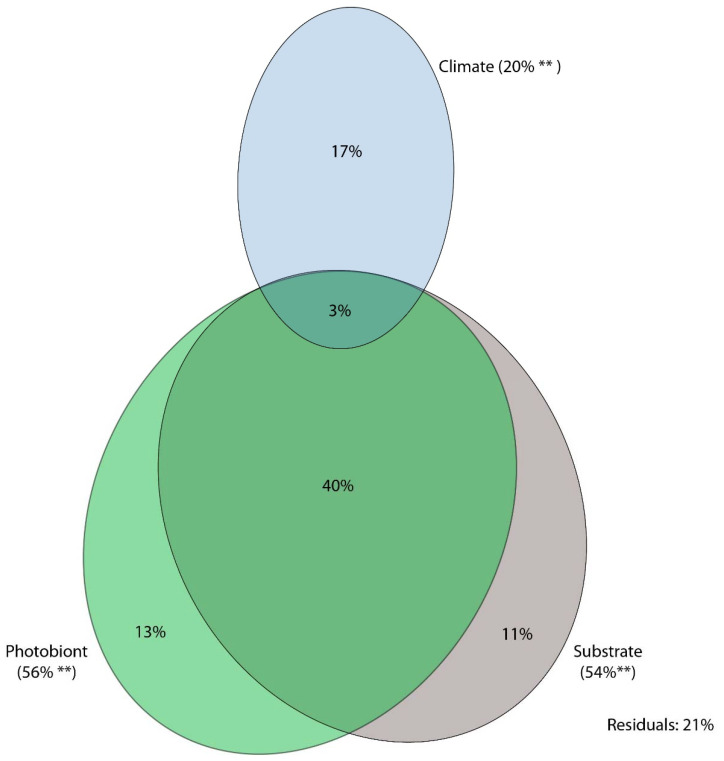
Results of variation partitioning, showing the percentage of mycobiont genetic distance explained based on three explanatory variables, i.e., climate, substrate, and photobiont genetic distance. Two asterisks (**) indicate significance at *p* < 0.01.

**Table 1 jof-08-00275-t001:** Characterisation of *Cladonia bellidiflora*, *C. polydactyla* and *C. umbricola* based on Ahti et al. [16] and James [19].

	*Cladonia bellidiflora*	*C. polydactyla*	*C. umbricola*
Morphology	Podetia yellowish green, 3–8 cm tall, usually ascyphose, little branched. Surface corticate, densely squamulose, never sorediate. Apothecia common, often large.	Podetia slender, pale grey to whitish or greenish grey, 1–3(–5) cm tall, unbranched or with few irregular branches, usually producing narrow scyphi. Surface of podetia sorediate (farinose to granulose), corticate or squamulose near the base. Apothecia infrequent.	Podetia pale greyish green or whitish grey, 1–3 cm tall, simple ascyphose or usually scyphose. Surface smooth, finely sorediate down to base. Apothecia rare.
Chemistry	Chemotype 1: usnic and squamatic acids (common throughout the world); chemotype 2: usnic and thamnolic acids (rarer: N and S America).	Thamnolic acid (rarely with small amounts of usnic acid).	Chemotype 1: squamatic acid; chemotype 2: thamnolic acid (rarer). In N America also other chemotypes with usnic and barbatic acids.
Habitat	Tundra, mountain heaths, humid rock outcrops, higher hilltops, stabilized scree.	On rotting wood and bases of trees, also on mossy rocks or soil.	On rotting wood and bases of trees in oceanic spruce forests. Usually in shade.
Distribution	Europe, Asia, North America, southern South America, New Zealand, subantarctic islands, Antarctica.	Western Europe, Macaronesia.	Western Europe (only Norway, British Isles and Spain), Macaronesia, western North and South America.

**Table 2 jof-08-00275-t002:** Comparison of the resolution of molecular markers used in this study.

Genetic Marker		No. of Variable Sites	No. of Parsimony Informative Sites	Intraspecific Distance	Interspecific Distance	Inter-/Intraspecific Distance
Mycobiont	ITS rDNA	14	7	0.0014	0.00207	1.48
	cox1	5	5	0.00104	0.00133	1.28
	EF-1α	2	2	0.00053	0.0016	3.04
	mtSSU	3	3	0.00031	0.00107	3.41
	RPB2	4	4	0.00047	0.00226	4.77
Photobiont	ITS rDNA	38	23	-	-	-

## Data Availability

The newly obtained sequences have been deposited in the GenBank database (accession numbers available in Appendix A).

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
