# Peer review of "An Exception to the Rule? Could Photobiont Identity Be a Better Predictor of Lichen Phenotype than Mycobiont Identity?"

_jof, 2022, doi:10.3390/jof8030275_

Round 1

Reviewer 1 Report

Dear Authors and Editor,

This is an exciting research area. There is much information about lichens, species concept,  speciation, and phenotypic differences in the literature, but this study is one of the new research which is going to test phenotypic differences by mycobiont/photobiont connection, by a molecular approach within three close Cladonia species.  Indeed the purpose of this paper is to discuss how to distinguish taxa that have distinct phenotypic differences, their photobiont diversity in connection to mycobiont diversity,  and potential factors among close species that best explain their phenotypic diversity and the emergence of symbiotic lichen phenotypes.  However, such a striking phenomenon warrants more in-depth coverage, critical assessments, and contextualization in this kind of study. 

The text is well written. In a few paragraphs, some degree of plagiarism has been recognized (I checked it via the Grammarly program). So those sentences must be checked and re-written.

corresponded better with lichen phenotype and separated esorediate C. bellidiflora from the two sorediate taxa. These results can be interpreted either as an unusual    taken from this paper:  https://onlinelibrary.wiley.com/doi/full/10.1110/ps.051397905

formed by the interaction of an exhabitant fungus and an extracellular arrangement of one or more photosynthetic partners and an indeterminate number of other microscopic “ taken from this paper: https://www.asianjournalofmycology.org/pdf/AJOM_3_1_18.pdf

Additionally, the text should explain why the study targets only one marker from the fungal part? Their explanation:   “Six molecular markers were amplified using the following primers: ITS1F [27] and ITS4 [28] for fungal ITS rDNA, CLRPB2-5F and CLRPB2-7R [21] for RPB2, mrSSU1 [29] and MSU7 [30] for mtSSU, cox1- 5959F and cox1-6711R [17] for the cox1 gene, CLEF-3F and CLEF-3R [21] for part of the elongation factor-1α, ITS1T and ITS4T [31] for algal ITS rDNA and actin_F and actin_R 130 [32] for the photobiont actin type I locus.”

Recently based on metagenomic data it was hypothesized that lichen species is some kind of dynamic biofilm.  Different degrees of the consortium of mycobiont, photobiont, lichen-associated yeasts, Endolichenic fungi, lichen-associated bacteria, and even lichenicolous fungi could change the Chemical profiles/ morphology since lichen thalli is a dynamic biofilm. 

Each species has its own chemical profile and morphological and phenotypic differences which may differ in different ecotypes or habitats.  Therefore, how would the authors connect this dynamic variation of Photobion/Mycobiont to the concept of lichen species? In some way, it must be explained.

Line 46 (crustose, microfilamentous or micro globose thalli). The lichenological thallus morphology concept (Foliose; Fruticose; Crustose.) must be followed. 

In this paper, it is not clear what is the Cladonia niche differentiation and what would be the Cladonia ecotype. It would be better if it could be expressed in one sentence. 

This study uses traditional chemical methods (TLC), so I recommend using LC-MS-based techniques for metabolite profiling of Cladonia species and incorporating the results into this study.

Author Response

Response to Reviewer 1

This is an exciting research area. There is much information about lichens, species concept,  speciation, and phenotypic differences in the literature, but this study is one of the new research which is going to test phenotypic differences by mycobiont/photobiont connection, by a molecular approach within three close Cladonia species.  Indeed the purpose of this paper is to discuss how to distinguish taxa that have distinct phenotypic differences, their photobiont diversity in connection to mycobiont diversity,  and potential factors among close species that best explain their phenotypic diversity and the emergence of symbiotic lichen phenotypes.  However, such a striking phenomenon warrants more in-depth coverage, critical assessments, and contextualization in this kind of study. 

  1. The text is well written. In a few paragraphs, some degree of plagiarism has been recognized (I checked it via the Grammarly program). So those sentences must be checked and re-written.
  • corresponded better with lichen phenotype and separated esorediate  bellidiflora from the two sorediate taxa. These results can be interpreted either as an unusual    taken from this paper:  https://onlinelibrary.wiley.com/doi/full/10.1110/ps.051397905
  • formed by the interaction of an exhabitant fungus and an extracellular arrangement of one or more photosynthetic partners and an indeterminate number of other microscopic “ taken from this paper: https://www.asianjournalofmycology.org/pdf/AJOM_3_1_18.pdf

The two mentioned sentences showed a random similarity with sentences from papers that focus on different subject (X-ray susceptibility of the lysine-pyridoxal-5′-phosphate Schiff base in Bacillus alcalophilus, and diversity of endolichenic fungi). We did neither read nor copy anything from these works.

  1. Additionally, the text should explain why the study targets only one marker from the fungal part? Their explanation:   “Six molecular markers were amplified using the following primers: ITS1F [27] and ITS4 [28] for fungal ITS rDNA, CLRPB2-5F and CLRPB2-7R [21] for RPB2, mrSSU1 [29] and MSU7 [30] for mtSSU, cox1- 5959F and cox1-6711R [17] for the cox1 gene, CLEF-3F and CLEF-3R [21] for part of the elongation factor-1α, ITS1T and ITS4T [31] for algal ITS rDNA and actin_F and actin_R 130 [32] for the photobiont actin type I locus.”

We studied five fungal markers (fungal ITS rDNA, mtSSU, cox1, RPB2 and elongation factor‐1α) and two algal markers (algal ITS rDNA and actin type I).

  1. Recently based on metagenomic data it was hypothesized that lichen species is some kind of dynamic biofilm.  Different degrees of the consortium of mycobiont, photobiont, lichen-associated yeasts, Endolichenic fungi, lichen-associated bacteria, and even lichenicolous fungi could change the Chemical profiles/ morphology since lichen thalli is a dynamic biofilm. 

We cite the studies that mention this phenomenon (eg. Spribille et al. 2016 and Spribille 2018) in the Introduction and Discussion.

  1. Each species has its own chemical profile and morphological and phenotypic differences which may differ in different ecotypes or habitats.  Therefore, how would the authors connect this dynamic variation of Photobion/Mycobiont to the concept of lichen species? In some way, it must be explained.

Thank you for this comment. This is true: there is a certain level of phenotypic variation depending on the environmental conditions. However, such a morphological variability is usually considered already in the description of the species, and the chemical variability is not related to the habitat. We have newly added Table 1 in which we summarized diagnostic morphological, chemical and ecological characteristics of the studied species.

  1. Line 46 (crustose, microfilamentous or micro globose thalli). The lichenological thallus morphology concept (Foliose; Fruticose; Crustose.) must be followed. 

We followed the terms there were used in the original publication that we cite (Honneger et al. 1998).

  1. In this paper, it is not clear what is the Cladonia niche differentiation and what would be the Cladonia ecotype. It would be better if it could be expressed in one sentence. 

Information about ecological preferences of Cladonia species studied have been newly included in Table 1.

  1. This study uses traditional chemical methods (TLC), so I recommend using LC-MS-based techniques for metabolite profiling of Cladonia species and incorporating the results into this study.

We do not think, that adding LS-MS data (in addition to our TLC data) into this study would change or improve our results. However, we may incorporate this technique into our future studies if it turns out to be beneficial.

Reviewer 2 Report

This is an excellent study providing important insight into potential speciation mechanisms and evidence for addressing recently evolving species complexes with incomplete lineage sorting. Overall, the manuscript is well written, perhaps with some minor style checks required. I have mostly some comments on the introduction and on how the discussion can be improved.

Line 50:

Traditionally, lichen phenotypes have been determined based on the primary mycobiont [3] ...

Suggest to rephrase; you likely mean that it has traditionally been assumed that the lichen phenotype is determined by the primary mycobiont, but as the phrase currently reads, it sound almost like it relates to taxonomy, i.e. how lichens are identified.

Line 58:

However, the most common examples of this phenomenon to date are the photosymbiodemes, in which the same fungus forms different structures, or entirely different lichens, when associating with either green algae or cyanobacteria.

This statement lacks examples and citations.

Line 71:

During our previous work, we observed that two distinctive Cladonia species (C. bellidiflora and C. polydactyla) that clearly differed morphologically, chemically [12], ecologically and in distribution range ...

Since you certainly intend a broad readership, most of your audience will have no access to this reference and so it is unclear what the actual differences are. I strongly recommend to add a table listing these. For instance, you do not mention that umbricola is intermediate between the two other species, morphologically similar to polydactyla but chemically more similar to bellidiflora. Therefore, the phrase "A third species (C. umbricola), close to C. polydactyla [12–14], was also included into the study to provide a more complete picture" hangs a bit in the air, although its inclusion in this study is basically mandatory.

Line 317-327:

Suggest to give examples here that have actually shown this, in addition to the theoretical papers cited. The most striking example is Usnea antarctica vs. U. aurantiacoatra, where largely the same markers as used here do not separate the two taxa but microsatellites and RADseq do (Grewe et al. 2018, Lagostina et al. 2018). You mention this in the next paragraph but you should perhaps separate the topic of incomplete lineage sorting, for which Usnea is an example, from the topic of difficult species delimitation in Cladonia, i.e. move the Usnea example to the previous pagagraph. Also check out Joel Mercado's work on the Cladonia sandstedei/subtenus complex, in his thesis and what he presented at the IAL [https://knowledge.uchicago.edu/record/3350]. His results would support your conclusions that this might be incomplete lineage sorting, rather than photobiont-mediated phenotype differentiation.

Regarding the latter, I am missing a discussion on how that could actually happen, i.e. why a particular photobiont strain would cause such a different morphology and chemistry. Are there studies about the isolated photobionts that suggest different physiologies? Are there other examples of lichens where the same photobiont strains are involved that suggest similar phenotypic differentiation in the resulting lichens? In relation to potentially photobiont-mediated phenotype, you should perhaps also discuss the concept of a developmental switch, which is either internal (e.g. photobiont) or external (environmental) triggers that determine a discrete phenotype early on. However, the incomplete lineage sorting hypothesis seems more likely and then one could interpret the photobiont specificity as avoiding photobiont competition?

In the discussion, I also suggest to go a bit more into detail regarding the differences in morphology, chemistry, ecology, and distribution. For instance, can bellidiflora vs. umbricola be considered a "species pair"? There are many examples of studies on apotheciate vs. sorediate species pairs, often showing phylogenetic differentiation, so this would be an interesting point to touch. Also, how chemically close are barbatic and thamnolic acid, i.e. how likely would be a switch between the two substances within the same mycobiont lineage? Perhaps give some examples of known variable chemistry in Cladonia involving the same substances. Notably, if you accept polydactyla/umbricola as chemical variants of a single species and then also apply the species pair concept, you could have a single, polymorphic lineage. This is not necessarily mutually exclusive with the incomplete lineage sorting argument, as one would envision an initially polymorphic lineage to eventually diverge into monomorphic sublineages during a speciation event.

Author Response

Response to Reviewer 2

This is an excellent study providing important insight into potential speciation mechanisms and evidence for addressing recently evolving species complexes with incomplete lineage sorting. Overall, the manuscript is well written, perhaps with some minor style checks required. I have mostly some comments on the introduction and on how the discussion can be improved.

  1. Line 50: Traditionally, lichen phenotypes have been determined based on the primary mycobiont [3] ...
    Suggest to rephrase; you likely mean that it has traditionally been assumed that the lichen phenotype is determined by the primary mycobiont, but as the phrase currently reads, it sound almost like it relates to taxonomy, i.e. how lichens are identified.

Thank you for this comment. We can see, that this formulation was not clear. We rephrased the sentence to “It has traditionally been assumed that the lichen phenotype is determined by the primary mycobiont [3] ...”

  1. Line 58: However, the most common examples of this phenomenon to date are the photosymbiodemes, in which the same fungus forms different structures, or entirely different lichens, when associating with either green algae or cyanobacteria.
    This statement lacks examples and citations.

We added citations to this statement. Other examples are present also in the Discussion section.

  1. Line 71: During our previous work, we observed that two distinctive Cladonia species (C. bellidiflora and C. polydactyla) that clearly differed morphologically, chemically [12], ecologically and in distribution range ...
    Since you certainly intend a broad readership, most of your audience will have no access to this reference and so it is unclear what the actual differences are. I strongly recommend to add a table listing these. For instance, you do not mention that umbricola is intermediate between the two other species, morphologically similar to polydactyla but chemically more similar to bellidiflora. Therefore, the phrase "A third species (C. umbricola), close to C. polydactyla [12–14], was also included into the study to provide a more complete picture" hangs a bit in the air, although its inclusion in this study is basically mandatory.

We agree. We added Table 1 summarizing morphological, chemical and ecological characteristics as well as distribution ranges of the three species. We also slightly rephrased the sentence that you mentioned.

  1. Line 317-327: Suggest to give examples here that have actually shown this, in addition to the theoretical papers cited. The most striking example is Usnea antarctica vs. U. aurantiacoatra, where largely the same markers as used here do not separate the two taxa but microsatellites and RADseq do (Grewe et al. 2018, Lagostina et al. 2018). You mention this in the next paragraph but you should perhaps separate the topic of incomplete lineage sorting, for which Usnea is an example, from the topic of difficult species delimitation in Cladonia, i.e. move the Usnea example to the previous pagagraph. Also check out Joel Mercado's work on the Cladonia sandstedei/subtenus complex, in his thesis and what he presented at the IAL [https://knowledge.uchicago.edu/record/3350]. His results would support your conclusions that this might be incomplete lineage sorting, rather than photobiont-mediated phenotype differentiation.

Thank you for this point. We added relevant sources to this paragraph.

  1. Regarding the latter, I am missing a discussion on how that could actually happen, i.e. why a particular photobiont strain would cause such a different morphology and chemistry. Are there studies about the isolated photobionts that suggest different physiologies? Are there other examples of lichens where the same photobiont strains are involved that suggest similar phenotypic differentiation in the resulting lichens? In relation to potentially photobiont-mediated phenotype, you should perhaps also discuss the concept of a developmental switch, which is either internal (e.g. photobiont) or external (environmental) triggers that determine a discrete phenotype early on. However, the incomplete lineage sorting hypothesis seems more likely and then one could interpret the photobiont specificity as avoiding photobiont competition?

We slightly rephrased a part of Discussion section in which we mention this. To our knowledge, there are no studies about different physiological performance of different photobiont strains, which would be relevant to our results.  We are only aware of two different Trebouxia strains (TR1 and TR9) with different physiological performace (for example Casano et al. 2011). However, these strains were coexisting in the thallus Ramalina farinacea, and thus this example is not applicable to our results. We don’t think that it is primarily the association with a particular photobiont, that triggers phenotype changes. Rather, we believe that a habitat differentiation is the main diversification force that is related both to photobiont selection (the mycobiont selects the photobionts which are the best adapted for the specific conditions) as well as to phenotypic differentiation. This we tried to describe on lines 376-390. Further in the text (lines 396-407), we also mention that diagnostic phenotypic features (mainly production of soredia) do not correspond consistently to association with a particular Asterochloris lineage in different groups of Cladonia. This is another hint suggesting that the morphological differentiation is not a direct consequence of association with a particular photobiont.

  1. In the discussion, I also suggest to go a bit more into detail regarding the differences in morphology, chemistry, ecology, and distribution. For instance, can bellidiflora vs. umbricola be considered a "species pair"? There are many examples of studies on apotheciate vs. sorediate species pairs, often showing phylogenetic differentiation, so this would be an interesting point to touch. Also, how chemically close are barbatic and thamnolic acid, i.e. how likely would be a switch between the two substances within the same mycobiont lineage? Perhaps give some examples of known variable chemistry in Cladonia involving the same substances. Notably, if you accept polydactyla/umbricola as chemical variants of a single species and then also apply the species pair concept, you could have a single, polymorphic lineage. This is not necessarily mutually exclusive with the incomplete lineage sorting argument, as one would envision an initially polymorphic lineage to eventually diverge into monomorphic sublineages during a speciation event.

We added information about the closeness of squamatic and thamnolic acids and about the common co-occurrence in other lichen genera (lines 354-357). Regarding species pair concept, we agree that C. bellidiflora and C. polydactyla/umbricola probably represent a species pair, but since we have no clear evidence for the incomplete lineage sorting hypothesis, we rather refrain from suggesting this option.

Reviewer 3 Report

Review on a manuscript for MDPI Journal of Fungi

An exception to the rule? Could photobiont identity be a better predictor of lichen phenotype than mycobiont identity?

by

Jana Steinová1*, Håkon Holien2, Alica Košuthová3, Pavel Škaloud1

The topic of the paper is very interesting. Lichen morphology is generally determined by the fungal partner and it is surprising to find an example within the group of podetiate Cladonias where a molecular evidence seems to break this rule. While photosymbiodemes were originally known as having green algae or cyanobacteria, the here presented examples are characterised by different species of the same photobiont genus Asterochloris.

The interpretation of the results originating from molecular genetic studies is not always clear in the taxonomy of Cladonia species. Analysis of genes characterising the photobiont might have an important contribution. This work studying the relation of 3 species presents direct relation between species and their Asterochloris species.

The manuscript is well written and illustrated. The English language text is all right.

Only some data should be added and a few minor mistakes are to be corrected.

The characterisation of the species is missing from the beginning of Materials and methods, though the following is written in the abstract

„two of three species clearly differ morphologically, chemically, ecologically and in distribution range” – please write more details on this, even if it is mentioned in your previous studies (cf. introduction) – perhaps a table summarising these features could be a solution.

Please, consider if you have not done so, that Smith et al. 2009 –referred as [58] mentions specimens with sorediate and squamulose podetia in C. polydactyla.

line 50

change „based on” to „on the basis of”

line 52             

The numbering of the references must be continuous, therefore change 7 and 6 and check it also later in the text and references.

line 77

change „to provide a more complete picture”

perhaps to „to have a wider view”

line 106

„… based on the results of previous studies focusing on the genus Cladonia.”

It is not clear from the sentence above if it refers to previous own studies or previous studies known from the literature, however from further text it seems so the you mean literature studies.

Thus adding sources „[16-21?]” is necessary to the end of this sentence.

line 109

add literature (any general or where it was used for Cladonia spp) source after  „…..ITS rDNA.” –

line 129

the colour of the entire line is darker than other lines in the mscr – check if it is boldface or different colour adjustment

line 196

add GenBank registration numbers here and to Supplement 1

The manuscript is suggested for publication after minor revision.

Author Response

Response to Reviewer 3

The topic of the paper is very interesting. Lichen morphology is generally determined by the fungal partner and it is surprising to find an example within the group of podetiate Cladonias where a molecular evidence seems to break this rule. While photosymbiodemes were originally known as having green algae or cyanobacteria, the here presented examples are characterised by different species of the same photobiont genus Asterochloris.

The interpretation of the results originating from molecular genetic studies is not always clear in the taxonomy of Cladonia species. Analysis of genes characterising the photobiont might have an important contribution. This work studying the relation of 3 species presents direct relation between species and their Asterochloris species.

The manuscript is well written and illustrated. The English language text is all right.

Only some data should be added and a few minor mistakes are to be corrected.

  1. The characterisation of the species is missing from the beginning of Materials and methods, though the following is written in the abstract: „two of three species clearly differ morphologically, chemically, ecologically and in distribution range” – please write more details on this, even if it is mentioned in your previous studies (cf. introduction) – perhaps a table summarising these features could be a solution.

We agree. We added Table 1 summarizing morphological, chemical and ecological characteristics as well as distribution ranges of the three species.

  1. Please, consider if you have not done so, that Smith et al. 2009 –referred as [58] mentions specimens with sorediate and squamulose podetia in C. polydactyla.

We included the information about squamulose character of the podetia near the base into the newly added Table 1.

  1. line 50: change „based on” to „on the basis of”

We rephrased this sentence because the meaning was not clear. Now it says: “It has traditionally been assumed that the lichen phenotype is determined by the primary mycobiont (3)….”

  1. line 52: The numbering of the references must be continuous, therefore change 7 and 6 and check it also later in the text and references.

Reference number 6 precedes reference number 7, and thus the numbering is correct. Please, check line 43.

  1. line 77: change „to provide a more complete picture”, perhaps to „to have a wider view”

Done as suggested.

  1. line 106: „… based on the results of previous studies focusing on the genus Cladonia.”
    It is not clear from the sentence above if it refers to previous own studies or previous studies known from the literature, however from further text it seems so the you mean literature studies.
    Thus adding sources „[16-21?]” is necessary to the end of this sentence.

Done as suggested. Sources have been added.

  1. line 109: add literature (any general or where it was used for Cladonia spp) source after  „…..ITS rDNA.”

Done as suggested.

8. line 129: the colour of the entire line is darker than other lines in the mscr – check if it is boldface or different colour adjustment

The format is identical with the rest of the text, and we cannot see (and correct) this (?).

  1. line 196: add GenBank registration numbers here and to Supplement 1

As agreed with the Editor, Genbank accession numbers will be added into the proofs.

The manuscript is suggested for publication after minor revision.